# Phylogenetic analysis of two single-copy nuclear genes revealed origin of tetraploid barley *Hordeum marinum*

**Bo Yin¹, Genlou Sun², Daokun Sun¹, Xifeng Ren**  **¹***

**1** College of Plant Science and Technology, Huazhong Agricultural University, Wuhan, China, **2** Biology Department, Saint Mary's University, Halifax, NS, Canada

* renxifeng@mail.hzau.edu.cn

**Data Availability Statement:** All relevant data are within the manuscript and its Supporting Information files.

**Funding:** Research supported by the Earmarked Fund for China Agriculture Research System

## Abstract

Sea barley *Hordeum marinum* is an important germplasm resource. However, the origin of this tetraploid *H. marinum* subsp. *gussoneanum* is still unclear, which has caused great perplexity to the exploration and utilization of germplasm resources. We used two single-copy nuclear genes, thioredoxin-like gene (*TRX*) and waxy1 gene encoding granule-bound starch synthase (*WAXY1*) to analyze 41 accessions of *Hordeum marinum*. The phylogenies of different genes told different story of evolution of tetraploids of *H. marinum* subsp. *gussoneanum*. The phylogenetic trees showed that two distinct copies of sequences from both genes were detected for some accessions of the tetraploids of *H. marinum* subsp. *gussoneanum*, and diploid *marinum* might also contribute to the origin and evolution of the tetraploid *gussoneanum*. Our findings suggested that tetraploid more likely originated from the diploids of *H. marinum* subsp. *gussoneanum* and another ancestor that might be an extinct unknown diploid species. Homogenization of gene in tetraploids also occurred after polyploidization as both *TRX* and *WAXY1* sequences in some accessions of tetraploid *H. marinum* subsp. *gussoneanum* cannot be distinguished, indicating the complicated evolution of this tetraploid.

## Introduction

The genus *Hordeum* in Triticeae consists of cultivated and wild barleys, including about 30 species with diploid (2n = 2x = 14), tetraploid (2n = 4x = 28) and hexaploid (2n = 6x = 42) taxa [1]. *Hordeum* species were proposed having four basic genome groups: I, H, Xa, and Xu [2]. Sea barley is a species in the Xa genome group [3], contains two subspecies: *H. marinum* subsp. *marinum* (2n = 2x = 14) and *gussoneanum* (2n = 2x = 14 and 2n = 4x = 28). The *marinum* and *gussoneanum* diploid forms can be distinguished by their morphology, but coexist throughout the Mediterranean region [3, 4]. The tetraploid *gussoneanum* is only found in the farthest eastern Mediterranean, from there to the east to Asia [4, 5]. Tetraploid subspecies was suggested as a suitable material for the improvement of barley in genetics and genomics, and investigation of polyploid evolution [6]. Over the last 50 years, many efforts have been made to clarify the relationships between tetraploid and diploid sea barley, while no phylogenic

(CARS-5). The Natural Sciences and Engineering Research Council of Canada (RGPIN-2018-05433) to GS.

**Competing interests:** The authors have declared that no competing interests exist.

consensus exists and the origin of the tetraploid sea barley of Xa-genome remains controversy [2, 5, 7–10].

The definition of genomes between Triticeae and many other crops often through cytogenetic characterization of chromosomes and the analysis of their chromosome pairing behavior between interspecific and intergeneric crosses [11]. Fluorescence *in situ* hybridization (FISH)-based karyotypes were used to clarify the relationships and the origin between their polyploids and diploids in Xa-genome species [3, 5, 6]. The autopolyploid origin of tetraploid *gussoneanum* was investigated by a series of cytogenetic methods, including C-banding karyotypes analysis and the meiotic behaviour in hybrid process [12], as well as molecular phylogenetic analysis from the chloroplast loci examination, geographical information and the ecological data [4]. Recent molecular phylogenetic analyses with single copy nuclear markers suggested allopolyploid origin of tetraploid *gussoneanum*, with Xa genome from diploid of *gussoneanum* and another genome from unclear diploid progenitor [3, 5, 13]. The *marinum* was also suggested as the second progenitor [2, 14], while other results indicated the second progenitor was an unclear extant or a probable extinct diploid of the *H. marinum* [13, 15].

*TRX* gene encodes a thioredoxin-like protein, which is a small-molecular-weight thermostable protein and plays an important role in plant growth and development. Orthologous genes have been isolated from many *Hordeum* species and the coding region is highly conserved among the *Hordeum* species [16]. The *TRX* gene was confirmed to be a single-copy nuclear gene and showed a high level of interspecific sequence polymorphisms. It has been used to explore the phylogeny of *Hordeum* species [14, 17, 18, 19]. The granule-bound starch synthase (*GBSSI* or *WAXY1*) gene is involved in the synthesis of starch granules and can produce the most abundant proteins related to starch granulocytes. It has been cloned from many plants including rice, maize, barley and wheat [20]. The *WAXY1* gene exists in a single copy in almost all the species studied and has been used to explore relationships among genera within tribes or among species within genera [21, 22, 23].

Here we used two single-copy nuclear genes: thioredoxin-like gene (*TRX*) and waxy1 gene encoding granule-bound starch synthase (*WAXY1*) to explore the origin of tetraploid species of *Hordeum marinum* subsp. *gussoneanum*.

## Materials and methods

### Plant materials

Six accessions of polyploid *H. marinum* subsp. *gussoneanum*, and 33 diploid species of *H. marinum* subsp. *marinum* and *H. marinum* subsp. *gussoneanum* were sequenced in our study, which are mainly distributed in the west coast of the Mediterranean Sea and the Middle East, including 4 accessions from Portugal, 11 from Spain, 12 from Greece and 3 from Bulgaria. While the tetraploid species are mainly distributed in the Middle East, such as Turkey, Iran and Afghanistan. Classification of *Hordeum* species/subspecies follows von Bothmer et al. [1]. The seeds were friendly provided by the Nordic Genetic Resource Center (NordGen). The detailed information of collector and donor for each species of *Hordeum* can be found from the Nordic Genetic Resource Center's Web site (www.nordgen.org/index.php/en/content/view/full/344). Seeds of each accession were germinated, transplanted and grown in a greenhouse. DNA was extracted from young leaf tissue collected from 5 to 10 plants of each accession using the method of Stein et al. [24]. Plant materials with accession number, genome constitution, country of origin, and GenBank identification numbers were given in Table 1. The sequences of other Triticeae species were downloaded from the NCBI database and included in the phylogenetic analysis, including *Aegilops*, *Psathyrostachys*, *Secale*, *Australopyrum*, *Taeniatherum*, *Triticum*, *Pseudoroegneria* and *Hordeum* (S1 Table).

**Table 1.  The species, accession number, genome and origin of 39 sea barley used in this study.**

| Species | Accession no. | Genome | Origin | *TRX* | *WAXY1* |
|---|---|---|---|---|---|
| H.*marinum* subsp. *marinum* Hudson | H41 | Xa | Turkey | + | + |
| | H56 | Xa | Germany | + | + |
| | H87 | Xa | Jordan | + | + |
| | H90 | Xa | Greece | + | + |
| | H109 | Xa | Greece | + | + |
| | H126 | Xa | Greece | + | + |
| | H155 | Xa | Greece | + | + |
| | H588 | Xa | Greece | + | + |
| | H607 | Xa | Greece | + | + |
| | H624 | Xa | Greece | + | + |
| | H631 | Xa | Greece | + | + |
| | H759 | Xa | Greece | + | + |
| | H158 | Xa | Portugal | + | + |
| | H508 | Xa | Spain | + | + |
| | H512 | Xa | Spain | + | + |
| | H515 | Xa | Spain | + | + |
| | H518 | Xa | Spain | + | + |
| | H524 | Xa | Spain | + | + |
| | H546 | Xa | Spain | + | + |
| | H559 | Xa | Spain | + | + |
| | H560 | Xa | Spain | + | + |
| | H568 | Xa | Spain | + | + |
| H. *marinum* subsp. *gussoneanum* (Parlatore) Thellung | H161 | Xa | Portugal | + | + |
| | H162 | Xa | Portugal | + | + |
| | H163 | Xa | Portugal | + | + |
| | H299 | Xa | Bulgaria | + | + |
| | H534 | Xa | Spain | + | + |
| | H563 | Xa | Spain | ++ | + |
| | H581 | Xa | Greece | + | + |
| | H598 | Xa | Greece | + | + |
| | H608 | Xa | Greece | + | + |
| | H823 | Xa | Bulgaria | + | + |
| | H837 | Xa | Bulgaria | + | + |
| H. *marinum* subsp. *gussoneanum* (Parlatore) Thellung | H64 | XaXa | Soviet Union | + | + |
| | H81 | XaXa | Afghanistan | + | + |
| | H208 | XaXa | Turkey | + | + |
| | H800 | XaXa | Iran | + | + |
| | H824 | XaXa | Iran | + | + |
| | H825 | XaXa | Turkey | + | + |

## DNA amplification and sequencing

The single copy nuclear gene *TRX* and *WAXY1* sequences were amplified by polymerase chain reaction (PCR) using the primers TRXF/R [17] and WAXY1F/R [21], respectively. The amplification profile for the *TRX* gene was as follows: an initial denaturation at 95˚C for 4 min; 14 cycles of 95˚C for 40 s denaturing and 72˚C for 90 s extension. The amplification profile for the *WAXY1* gene was as follows: an initial denaturation at 94˚C for 3 min; 35 cycles of 94˚C for 30 s, 65˚C for 40 s, 72˚C for 1 min; and a final cycle of 72˚C for 20 min. PCR was carried

out in a 20 μL reaction mixture, containing 30 ng template DNA, 0.2μM of each primer, 1.5 mM MgCl2, 0.2 mM of each deoxynucleotide (dATP, dCTP, dGTP, dTTP), 1 U of high fidelity *Taq* DNA polymerase (Biolabs, New England), and distilled deionized water to the final volume.

To enhance the sequence quality, each PCR product from the two nuclear genes was independently amplified twice, and cloned into pGEMeasy T vector (Promega Corporation, Madison, Wis., USA) following the manufacturer's instruction. At least 10 clones from each species were randomly selected. In order to detect the presence of an insert, each colony was firstly transferred to 10 μL of LB broth with 0.1 mg/mL Ampicillin antibiotics. These solutions were incubated at 37˚C for 30 min, then 2 μL was used as template to check the presence of an insert by PCR using the initial primers. For the colonies that were detected to contain the insert, the remaining 8 μL of solution was transferred to another 5 mL of LB broth (with Ampicillin antibiotics) and incubated at 37˚C overnight for plasmid DNA isolation. Sequencing was done commercially by the Taihe Biotechnology (Beijing, China).

## Data analysis

Automated sequence outputs were visually inspected using chromatographs. Multiple sequence alignments were made using ClustalX using default parameters [25]. We checked the existence of recombinants in our research by closer inspection of sequence alignment before phylogenetic analysis, and no such recombinants were found. A phylogenetic tree of sequences was constructed using the maximum likelihood (ML) algorithm in MEGA7, with 1000 bootstrap replicates, and a 60% cut off was used in the analysis. All characters were specified as unweighted and unordered, and all gap-only columns were excluded in the analyses. The most parsimonious trees were obtained by performing a heuristic search using the Tree Bisection-Reconnection (TBR) with the following parameters: Mul-Trees on and 10 replications of random addition sequences with the stepwise addition option. A strict consensus tree was usually generated from multiple parsimonious trees. Overall character congruence was estimated by the consistency index (CI) and the retention index (RI). Bootstrap (BS) values with 1000 replications were calculated by performing a heuristic search using the TBR option with Multree on, and were used to infer the robustness of clades [26].

In addition, Bayesian analyses were performed using MrBayes 3.1 [27]. Each data was tested to find the best-fitting model of sequence evolution by jModelTest 2.1.10 [28], using default parameters. Each data was estimated by the Akaike information criterion (AIC) [29], Bayesian information criterion (BIC) [30], and a decision theoretic performance-based approach (DT) [31]. The BIC was used to select model because of its high accuracy [28]. The HKY+G and TrN+G+I substitution models led to the best BIC scores for TRX and WAXY1, respectively (S2 Table). Therefore, the HKY+G (for TRX) and TrN+G+I (for WAXY1) models were used in the Bayesian analysis using MrBayes 3.1 [27]. The program was run with the standard setting of two analyses in parallel, each with four chains, and estimates of convergence of results by calculating standard deviation of split frequencies between analyses. In total, 3040 000 generations for TRX and 835000 generations for WAXY1 were run to make the standard deviation of split frequencies below 0.01. Samples were taken every 1000 generations. For all analyses, the first 25% of samples from each run were discarded as burn-in to ensure the stationary of the chains. A majority rule consensus tree was generated from the remaining sampled trees. Bayesian posterior probability (PP) values were obtained, and were used to test the robustness of clades.

**Table 2. Estimates of nucleotide diversity per base pair and test statistics for selection at *TRX* gene in different barley populations.** The three populations are T-g-4x (*H. marinum* subsp. *gussoneanum4x*), T-m-2x (*H. marinum* subsp. *marinum* 2x) and T-g-2x (*H. marinum* subsp. *gussoneanum* 2x) respectively. Note: the gaps/missing/data were excluded; *significant at 0.05 level.

| Population | No. of accessions | No. of Haplotypes (H) | Haplotyp diversity (Hd) | Number of polymorphic sites (S) | Theta (per site) from S (θ) | Nucleotid diversity (π) | Tajima's D test | Fu and Li's D* test | Fu and Li's F* test |
|---|---|---|---|---|---|---|---|---|---|
| T-g-4x | 6 | 14 | 0.971 | 120 | 0.01466 ±0.00372 | 0.05071 | 0.90768 | 0.84171 | 0.99820 |
| T-m-2x | 22 | 4 | 0.671 | 5 | 0.00075 ±0.00063 | 0.00246 | 2.21537* | 1.17564 | 1.69999* |
| T-g-2x | 11 | 1 | 0.000 | 0 | 0.00000 ±0.00000 | 0.00000 | 0.00000 | 0.00000 | 0.00000 |
| All | 39 | 18 | 0.881 | 125 | 0.03215 ±0.00922 | 0.03089 | -0.24721 | 0.68653 | 0.40200 |

## Results

### Genetic diversity analysis and neutrality test

To reveal the genetic diversity of *TRX* between the diploid *H. marinum* subsp. *gussoneanum* and the diploid *H. marinum* subsp. *marinum*, genetic analysis and neutrality test in three populations were performed (Table 2). The number of haplotypes (H = 4), the haplotype diversity (Hd = 0.671), the number of polymorphic sites (S = 5), the per-site nucleotide diversity (θ = 0.00075 ± 0.00063) and the nucleotide diversity (π = 0.00246) were observed in the diploid *H. marinum* subsp. *marinum*. While just one haplotype was observed in the diploid *H. marinum* subsp. *gussoneanum* population.

In addition to these, Tajima and Fu and Li's tests also were measured in three populations. Values obtained from three populations all are positive. However, only diploid species of *H. marinum* subsp. *marinum* population showed significant Tajima and Fu and Li's F values (2.21537 and 1.69999, respectively).

As shown in Table 3 for *WAXY1*, the number of haplotypes (H = 5), the haplotype diversity (Hd = 0.727), the number of polymorphic sites (S = 15), the per-site nucleotide diversity (θ = 0.00222 ± 0.00140) and the nucleotide diversity (π = 0.00955) were observed in the diploid species of *H. marinum* subsp. *marinum* population. The number of haplotypes (H = 3), the haplotype diversity (Hd = 0.709), the number of polymorphic sites (S = 2), the per-site nucleotide diversity (θ = 0.00089 ± 0.00068) and the nucleotide diversity (π = 0.00114) were observed in the diploid species of *H. marinum* subsp. *gussoneanum* population. Compared with the diploid *H. marinum* subsp. *marinum* population, 1.8% haplotype diversity (Hd) and 0.841%

**Table 3. Estimates of nucleotide diversity per base pair and test statistics for selection at *WAXY1* gene in different barley populations.** The three populations are W-g-4x (*H. marinum* subsp. *Gussoneanum* 4x), W-m-2x (*H. marinum* subsp. *marinum* 2x) and W-g-2x (*H. marinum* subsp. *gussoneanum* 2x), respectively. Note: the gaps/missing/data were excluded; *significant at 0.05 level.

| Population | No. of accessions | No. of Haplotypes (H) | Haplotyp diversity (Hd) | Number of polymorphic sites (S) | Theta (per site) from S (θ) | Nucleotid diversity (π) | Tajima's D test | Fu and Li's D* test | Fu and Li's F* test |
|---|---|---|---|---|---|---|---|---|---|
| W-g-4x | 6 | 17 | 0.993 | 58 | 0.00813 ±0.00292 | 0.02781 | 0.70277 | -0.35238 | -0.05040 |
| W-m-2x | 22 | 5 | 0.727 | 15 | 0.00222 ±0.00140 | 0.00955 | 2.73675** | 1.52352** | 2.19348** |
| W-g-2x | 11 | 3 | 0.709 | 2 | 0.00089 ±0.00068 | 0.00114 | 0.85048 | 0.99697 | 1.07842 |
| All | 39 | 16 | 0.901 | 65 | 0.01961 ±0.00596 | 0.02371 | 0.49478 | 0.26653 | 0.41601 |

nucleotide diversity (π) reduction were found in the diploid *H. marinum* subsp. *gussoneanum* population.

Both Tajima's D, and Fu and Li's tests were positive for the diploid *H. marinum* subsp. *marinum* population and the diploid *H. marinum* subsp. *gussoneanum* population, while negative for the tetraploid species of *H. marinum* subsp. *gussoneanum* population in Fu and Li's tests. Tajima's D and Fu and Li's values were observed significantly for the diploid *H. marinum* subsp. *marinum* population. However, Tajima's D and Fu and Li's tests did not show significantly for other two.

## Phylogenetic analysis of *TRX* sequence

The DNA were amplified and cloned from 6 accessions of polyploid species and 33 diploid species. A total of 69 sequences (45 sequences generated in this study, and 24 downloaded from GenBank) were used for phylogenetic analysis. Two distinct copies of sequences were recovered from each tetraploid *gussoneanum* accession. The 19 *TRX* sequences from *H. marinum* subsp. *gussoneanum* and *H. murinum* subsp. *murinum* and 49 *TRX* sequences from other diploid *Hordeum* species together with one *TRX* sequence from the specie of *Psathyrostachys juncea* were phylogenetically analyzed. *Psathyrostachys juncea* was used as the outgroup. The data matrix contained 1157 characters, of which 622 were constant, 204 were parsimony uninformative, and 331 were parsimony informative. Maximum parsimony analysis produced 645 equally parsimonious trees with CI = 0.775 and RI = 0.894 (excluding uninformative characters). The separated Bayesian analyses using the HKY+G model led to the mean log-likelihood values of identical trees is –5772.39 and –5707.94. The minor difference between the tree topologies in Bayesian trees and those generated by MP was observed. The consensus tree with PP value produced by Bayesian analysis was shown in Fig 1.

Phylogenetic analyses based on *TRX* sequence data grouped most sequences from the Xa-genome species of *H. marinum* subsp. *gussoneanum* and *H. marinum* subsp. *marinum* into a clade with 0.73 PP support (BS = 100%), All sequences from diploid species of *H. marinum* subsp. *marinum* formed a clade with 0.85 PP support (BS = 97%), and there are two subclades with PP = 0.96 (BS = 98%) and PP = 0.86 (BS = 96%) internally. All diploid species of *H. marinum* subsp. *gussoneanum* formed a clade in 0.76 PP support (BS = 100%). One each copy from 6 tetraploid species of *H. marinum* subsp. *gussoneanum* was grouped closely with diploid species of *H. marinum* subsp. *gussoneanum* and *H. marinum* subsp. *marinum*, while sequences from another copy of each 6 tetraploid *H. marinum* subsp. *gussoneanum* were grouped into an independent clade with PP = 0.98 (BS = 100%) (Fig 1).

## Phylogenetic analysis of *WAXY1* sequence

A total of 76 *WAXY1* sequences from 39 accessions, including 6 polyploid *H. marinum* and 33 diploid species of *H. marinum* and other diploid species in Triticeae downloaded from GenBank, were obtained. *Psathyrostachys juncea* was used as the outgroup in phylogenetic analysis. The data matrix contained 1299 characters, of which 871 were constant, 159 were parsimony uninformative, and 269 were parsimony informative. Maximum parsimony analysis produced 173 equally parsimonious trees with CI = 0.624 and RI = 0.912 (excluding uninformative characters). The separated Bayesian analyses using the TrN+G+I model led to the mean log-likelihood values of identical trees was –6036.17 and –5979.37. The minor difference between the tree topologies in Bayesian trees and those generated by MP was observed. The consensus tree with PP value produced by Bayesian analysis was shown in Fig 2.

Phylogenetic analyses based on *WAXY1* sequence data grouped all sequences from the Xa-genome species with 0.86 PP support (BS = 63%) (Fig 2). All the diploid *H. marinum* subsp.

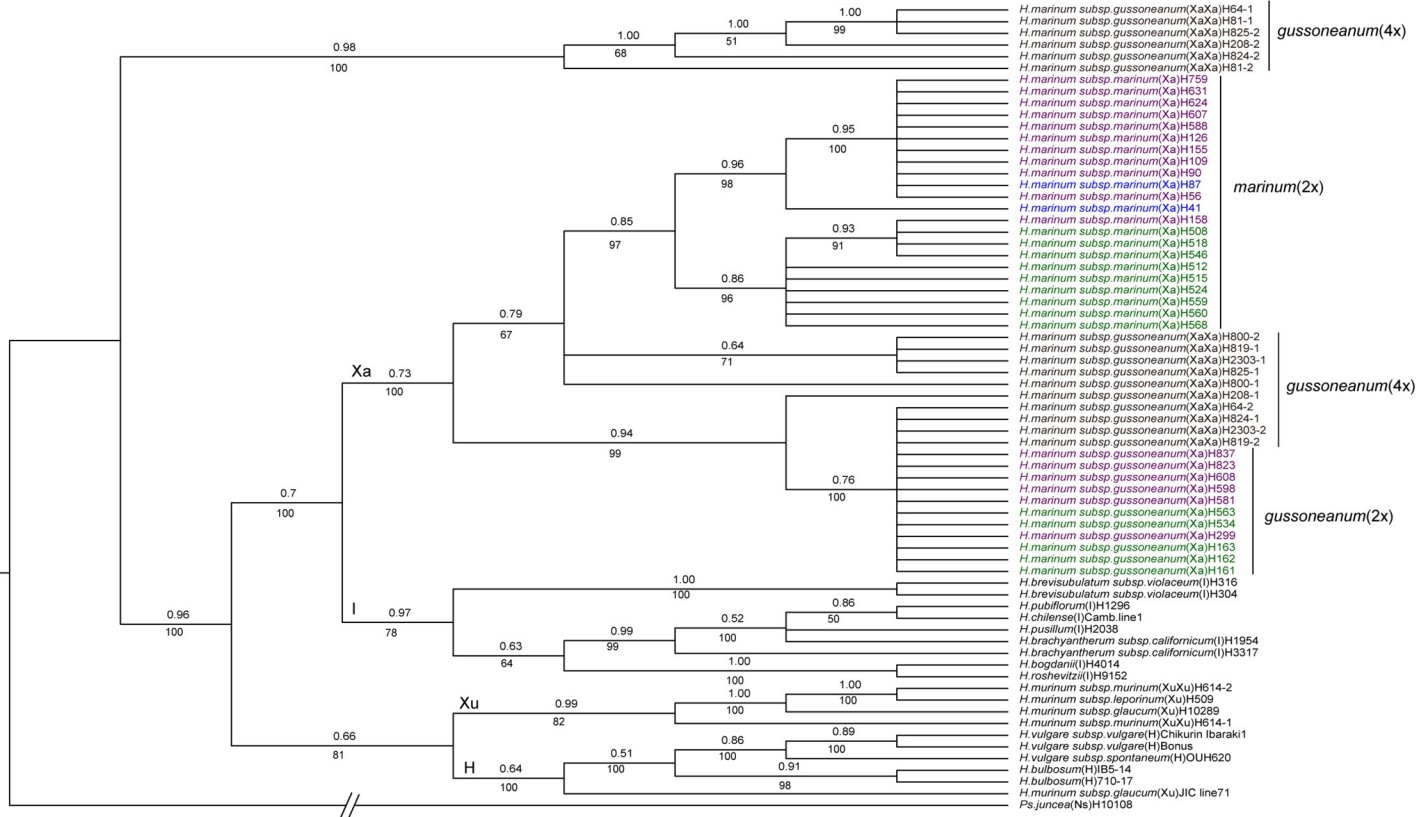

**Fig 1. Strict consensus trees derived from *TRX* sequence data was conducted using the HKY+G model by Bayesian analysis.** The main topologies generated by Maximum parsimony analysis are similar. Numbers above and below branches are Bayesian posterior probability (PP) values and bootstrap values, respectively. *Psathyrostachys juncea* was used as the outgroup. Consistency index (CI) = 0.775, retention index (RI) = 0.894. Geographic distribution of species: western Mediterranean (green), central Mediterranean (purple), eastern Mediterranean (blue).

*gussoneanum* and *H. marinum* subsp. *marinum* were grouped into a clade with 1.00 PP support (BS = 100%), and sequences from diploid *H. marinum* subsp. *marinum* formed two subclades with 1.00 PP support (BS = 99%) and 0.56 PP support (BS = 91%). One each copy from 6 tetraploid *H. marinum* subsp. *gussoneanum* formed a clade with 1.00 PP support (BS = 93%). The rest sequences from another copy of 6 tetraploid *H. marinum* subsp. *gussoneanum* and all sequences from diploid *H. marinum* subsp. *gussoneanum* formed a subclade with 1.00 PP support (BS = 84%), while five of the sequences from diploid *H. marinum* subsp. *gussoneanum* were grouped in one clade with 0.55 PP support (BS = 74%), included in this clade was the sequence H64-1 from tetraploid *H. marinum* subsp. *gussoneanum* (Fig 2).

## Discussion

### Origin of polyploids in tetaploid *H. marinum*

The relationships within the Xa genome have received considerable attention, partly on account of the importance of germplasm resources in barley, and partly because many previous phylogenetic research data sets have failed to reach a consensus view of the relationships among the Xa genome [32]. Phylogenetic relationships within Xa genome species are still unclear. Polyploid species present numerous opportunities and challenges to molecular phylogenetic studies. The maternally-inherited chloroplast genome data of polyploid is generally

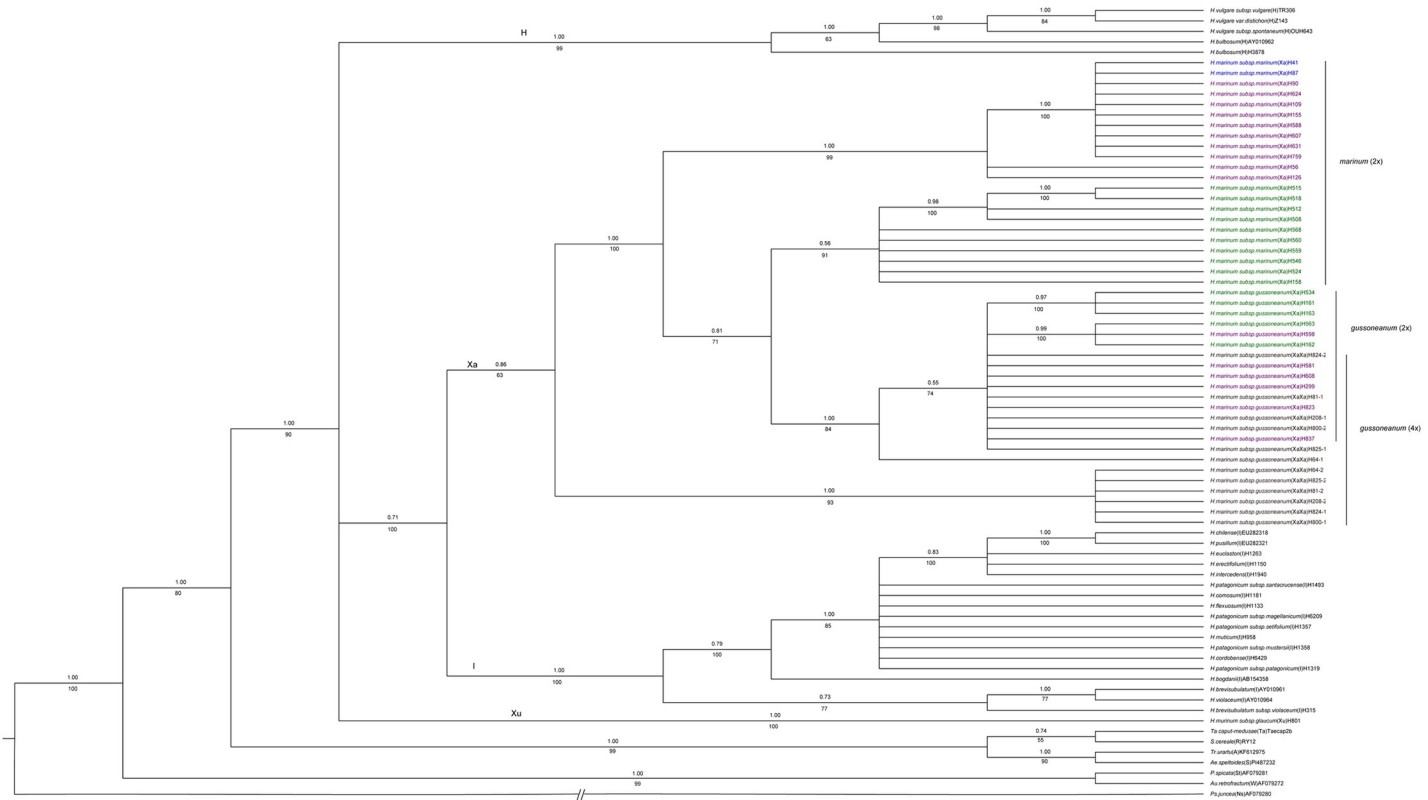

**Fig 2. Strict consensus trees derived from *WAXY1* sequence data was conducted using the TrN+G+I model by Bayesian analysis.** The main topologies generated by Maximum parsimony analysis are similar. Numbers above and below branches are Bayesian posterior probability (PP) values and bootstrap values, respectively. *Psathyrostachys juncea* was used as the outgroup. Consistency index (CI) = 0.624, retention index (RI) = 0.912. Geographic distribution of species: western Mediterranean (green), central Mediterranean (purple), eastern Mediterranean (blue).

easy to obtain, and they provide more potentially informative data but many are incomplete. While data from single-copy or low-copy markers is especially informative [4, 33]. Phylogenies *Hordeum* species including Xa genome species have been previously reported using *TRX* and *RPB2* sequences [14, 34]. While the origin and evolution of Xa-genome tetraploids has been controversy. Although homologous origin hypothesis [4, 12] or heterologous origin hypothesis [2, 13, 14, 15] have been proposed in previous studies, no enough evidence and data confirm that *H. marinum* subsp. *gussoneanum* is autopolyploids or allopolyploids. Therefore, we used two single-copy nuclear genes *TRX* and *WAXY1* to infer phylogenetic relationships between diploid species and tetraploid species of *H. marinum* species.

The phylogenetic tree of *TRX* showed that the accessions of tetraploid *H. marinum* subsp. *gussoneanum* such as H81, H800, H819 and H2303 are more likely autopolyploid. Two copies of sequences in each these accessions were placed into closely related the sister sub-clade, indicating that the two genomes in each these accessions might be less differentiated, which makes we speculate that they have a homologous origin. While the two copies of sequences from the accessions of tetraploid *H. marinum* subsp. *gussoneanum* such as H64, H208, H824 and H825 were placed into distinct clades, showing that their two genomes are quite different. We speculated that the genomes in each these accessions might come from different sources. The phylogenetic tree of *TRX* showed that H64, H208, H824 and H825 are heterologous origins, H819, H2303, H81, and H800 are likely homologous origins. While the phylogenetic tree of *WAXY1* showed that H64, H81, H208, H800, H824, and H825 are all heterogeneous origins. The reason

for this phenomenon might be due to different genes having different evolutionary history, the high degree of differentiation of the *TRX* gene and the large variability in different species of the *Triticeae* have occurred [14, 19]. One each copy of sequences from H800, H819 and H2303 is close to the diploid *H. marinum* subsp. *gussoneanum* and another each copy is close to the diploid *H. marinum* subsp. *marinum* in the phylogenetic tree of *TRX*, while only diploid *H. marinum* subsp. g*ussoneanum* and tetraploid *H. marinum* subsp.*gussoneanum* were grouped in the same subclade and the diploid *H. marinum* subsp. *marinum* was grouped in a subclade alone in the phylogenetic tree of *WAXY1*. Results suggested that diploid *H. marinum* subsp. g*ussoneanum* and tetraploids are closely related and share partially similar gene sequences, which suggests the diploid *H. marinum* subsp. *gussoneanum* might be one of the ancestors of the tetraploid *H. marinum* subsp. *gussoneanum*. This was consistent with the putative involvement of diploids *H. marinum* subsp. *gussoneanum* as a diploid ancestor of tetraploids of *H. marinum* subsp. g*ussoneanum* [3]. Similarly, genomic *in situ* hybridization (GISH) revealed the diploid forms of *gussoneanum* would be a maternal diploid ancestor of *H. marinum* tetraploids [3], as provided by molecular phylogenies based on chloroplast DNA [4, 35]. While some tetraploids of *H. marinum* subsp. g*ussoneanum* and diploids of *H. marinum* subsp. *marinum* were grouped in a sister-subclade, indicating diploid *marinum* might also contribute to the origin and evolution of the tetraploid *gussoneanum* [3]. Our results also suggested that the second genome donor to tetraploid *gussoneanum* remains unknown, and it might be an extinct species. Based on the two nuclear gene sequence data, we still could not determine the second genome donor to tetraploid *gussoneanum* since only tetraploid accession H824 was grouped into one cluster with diploid for both genes, other tetraploid accessions changed when they were analyzed with two distinct genes. These results suggested that different genes might experience different evolutionary history. In order to detangle the origin of tetraploid *gussoneanum*, the more genes used in the study, the better results will be obtained.

Many polyploid species have likely originated repeatedly, including genetically different parent species. This supports the multiple origin hypothesis of the parents [11]. The heterogeneous origin can also be seen from the geographical distribution of diploids and tetraploids. Our findings indicated that the tetraploid *gussoneanum* might be originated in two or more geographic origin center [3]. Some studies indicated that tetraploids are mainly distributed in the Middle East and Eurasia, and some appeared in North America. While diploids are widely distributed along the Mediterranean coast as well as in Portugal and Spain. Considering the important impact of climate changing on survival and evolution of species, we speculate that the extinction of another ancestor might be related to climate change. Previous studies also mentioned that the climate or living environment is not suitable for the ancestors of the tetraploid *H. marinum* subsp. *gussoneanum* growth as the glaciers melting [36, 37], and it became extinct after the formation of tetraploids of *H. marinum* subsp. *gussoneanum*, resulting in a genetic bottleneck and the loss of some inherited parental information [4, 38].

In summary, Our findings favor heterogeneous origin of the tetraploid *H. marinum* subsp. *gussoneanum*. Homogenization of gene in tetraploids also occurred after polyploidization as both *TRX* and *WAXY1* sequences in some accessions of tetraploid *H. marinum* subsp. *gussoneanum* cannot be distinguished.

## Supporting information

**S1 Table. Taxa from *Aegilops*, *Psathyrostachys*, *Secale*, *Taeniatherum*, *Australopyrum*, *Triticum*, *Pseudoroegneria* and *Hordeum* used in this study.**
(DOC)

**S2 Table. Maximum likelihood fits of nucleotide substitution models.**
(DOC)

## Author Contributions

**Conceptualization:** Genlou Sun, Xifeng Ren.

**Data curation:** Bo Yin, Xifeng Ren.

**Formal analysis:** Xifeng Ren.

**Funding acquisition:** Xifeng Ren.

**Investigation:** Xifeng Ren.

**Methodology:** Bo Yin.

**Project administration:** Genlou Sun.

**Resources:** Genlou Sun, Daokun Sun.

**Software:** Bo Yin.

**Supervision:** Daokun Sun.

**Validation:** Bo Yin.

**Writing – original draft:** Bo Yin, Xifeng Ren.

**Writing – review & editing:** Genlou Sun, Daokun Sun.

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
