## [Decision Letter · Decision Letter 0]

28 May 2020

PONE-D-20-12057

Phylogenetic analysis of two single-copy nuclear genes revealed origin of tetraploid barley Hordeum marinum

PLOS ONE

Dear Dr. Ren,

Thank you for submitting your manuscript to PLOS ONE. After careful consideration, we feel that it has merit but does not fully meet PLOS ONE’s publication criteria as it currently stands. Therefore, we invite you to submit a revised version of the manuscript that addresses the points raised during the review process.

Please seek a professional proofreading service to improve the English;Address all comments / concerns raised by the two reviewers;

We look forward to receiving your revised manuscript.

Kind regards,

Wujun Ma

Academic Editor

PLOS ONE

Journal Requirements:

Reviewers' comments:

Reviewer's Responses to Questions

**Comments to the Author**

1. Is the manuscript technically sound, and do the data support the conclusions?

Reviewer #1: Yes

Reviewer #2: Yes

2. Has the statistical analysis been performed appropriately and rigorously? 

Reviewer #1: Yes

Reviewer #2: Yes

3. Have the authors made all data underlying the findings in their manuscript fully available?

Reviewer #1: Yes

Reviewer #2: Yes

4. Is the manuscript presented in an intelligible fashion and written in standard English?

Reviewer #1: Yes

Reviewer #2: Yes

5. Review Comments to the Author

Reviewer #1: The study used two single-copy nuclear gene markers to classify a collection of 33 Hordeum marinum species, in order to draw a phylogenetic consensus map and reveal the origin of the tetraploid subspecies in Hordeum marinum. #### If my synopsis of the article is wrong, please discard my comments and find an alternative reviewer.######

It is a potentially article because of the novelty of the work and new findings revealed by one of the gene marker derived from the TRX gene. Compared to previous work [refer to reference number 3], the study has found similar results that the gussoneanum (4x) was clustered into two clades by the TRX gene marker, one of the clade was grouped with the Xa genome, but the other clade was not grouped into any genome. In comparison, by using the WAXY1 gene marker, all of the gussoneanum (4x) were grouped together, but overlapped with some accessions from the gussoneanum (2x) group. My question is: is that possible the unclustered clade may carry a new genome that was never reported before, or the overlapped group the gussoneanum (4x) and gussoneanum (2x) accessions is the transition stage from the gussoneanum (2x) type evolving to the gussoneanum (4x) type?

By reading the manuscript, the answer seems unclear.

Comments on other parts of the manuscript:

1. It is unclear why nuclear TRX and WAXY1 gene markers were selected out of numerous other genes. Authors must have very sound reasons to do so. As a result, the reasons should be elaborated in the introduction part. Why they are selected? If they have been used for classification before, what are the main findings? If they were not used before, what's the advantages of using them and expected outcomes?

2. There are a few places where the font of the words are different from the rest. I suggest using line numbers on pages to identify sentences when resubmit.

3. Reference 25: format

4. Fig.1 and Fig.2 sentence "Geographic distribution of ...." sentence is incomplete, the denotation for the "red line" is missing.

5. The last paragraph or "conclusion" paragraph needs to rewrite. This part should only have results from the study and concluding remarks from the authors, so the citation is redundant. Again, the word "angiosperms"

is inappropriate in presenting here.

Reviewer #2: This paper was written very well. Sea barley Hordeum marinum is an important germplasm resource. This paper suggested that tetraploid more likely originated from the diploids of H. marinum subsp. gussoneanum and the other ancestor that might be an extinct unknown diploid species. Homogenization of gene in tetraploids also occurred after polyploidization as both TRX and WAXY1 sequences in some accessions of tetraploid H. marinum subsp. gussoneanum cannot be distinguished, indicating the complicated evolution of this tetraploid. Line H824 was an interesting line, the gene was grouped into one cluster with 2X for both genes. But other 4X lines changed when they were analysed with two distinct genes. It means that the more genes used in the study, the better results will be obtained. Discuss this in the paper.

There are some other minor issues which need to be addressed.

Page 8:

Abstract: "H. marinum subsp. gussoneanum, diploid marinum" from two sentences, insert “and”.

P13: Rewrite this sentence “To reveal the genetic differentiation between the diploid H. marinum subsp. Gussoneanum and the diploid H. marinum subsp. marinum used TRX, genetic analysis and neutrality test in three populations were performed.”

P13: change 1 from “While just 1 haplotype” to one. 1 to 9 should be written out.

Reference:

Italic Hordeum marinum for the reference 4. Double check other references.

6. PLOS authors have the option to publish the peer review history of their article (what does this mean?). If published, this will include your full peer review and any attached files.

Reviewer #1: No

Reviewer #2: No

---

## [Author Response · Author response to Decision Letter 0]

15 Jun 2020

June 15, 2020

Dear Wujun Ma:

Thanks for providing us with an opportunity to revise our manuscript. We have taken all comments from reviewers into consideration, and made a revised version of this manuscript. We invited Professor Genlou Sun (Saint Mary University of Canada) and Professor Hong Luo (Clemson University) to revise this manuscript. I am sending you our revised manuscript along with a point-by-point description of the changes made for your reviewing. I hope that our manuscript will meet your standards for publication in Plos one.

Sincerely yours,

Xifeng Ren

Description of the changes 

Journal Requirements:

Response: Thanks for your suggestion. Changes were made according to the suggestions.

Response: Thanks for your suggestion. Changes were made according to the suggestions. We have carefully edited this manuscript.

Reviewer #1

Reviewer #1: The study used two single-copy nuclear gene markers to classify a collection of 33 Hordeum marinum species, in order to draw a phylogenetic consensus map and reveal the origin of the tetraploid subspecies in Hordeum marinum. If my synopsis of the article is wrong, please discard my comments and find an alternative reviewer.

It is a potentially article because of the novelty of the work and new findings revealed by one of the gene marker derived from the TRX gene. Compared to previous work [refer to reference number 3], the study has found similar results that the gussoneanum (4x) was clustered into two clades by the TRX gene marker, one of the clade was grouped with the Xa genome, but the other clade was not grouped into any genome. In comparison, by using the WAXY1 gene marker, all of the gussoneanum (4x) were grouped together, but overlapped with some accessions from the gussoneanum (2x) group. My question is: is that possible the unclustered clade may carry a new genome that was never reported before, or the overlapped group the gussoneanum (4x) and gussoneanum (2x) accessions is the transition stage from the gussoneanum (2x) type evolving to the gussoneanum (4x) type?

By reading the manuscript, the answer seems unclear.

Response: Thanks for your suggestion. Because the limited number of genes used in the experiment, the origin of tetraploids of H. marinum subsp. gussoneanum has not been fully explored. Yes, the unclustered clade may carry a new genome that was never reported before. It may come from an extinct species. Similar results have appeared in previous literature reports.

Comments on other parts of the manuscript:

1. It is unclear why nuclear TRX and WAXY1 gene markers were selected out of numerous other genes. Authors must have very sound reasons to do so. As a result, the reasons should be elaborated in the introduction part. Why they are selected? If they have been used for classification before, what are the main findings? If they were not used before, what's the advantages of using them and expected outcomes?

Response: Thanks for your comments. We have added this information in the introduction. The nuclear TRX and WAXY1 genes were selected as markers because they are relatively conservative during the evolution of Triticeae, and they have also been used in phylogenetic studies.

2. There are a few places where the font of the words are different from the rest. I suggest using line numbers on pages to identify sentences when resubmit.

Response: Thanks for your suggestion. Changes were made according to the suggestions. 

3. Reference 25: format

Response: Thanks for your suggestion. We have corrected the format error of the reference. 

4. Fig.1 and Fig.2 sentence "Geographic distribution of ...." sentence is incomplete, the denotation for the "red line" is missing.

Response: Red is only labeled as tetraploids of H. marinum subsp. gussoneanum, and its geographical distribution is shown in Table 1. For readability, we have changed the red label to black, with pictures and notes. 

5. The last paragraph or "conclusion" paragraph needs to rewrite. This part should only have results from the study and concluding remarks from the authors, so the citation is redundant. Again, the word "angiosperms"

is inappropriate in presenting here.

Response: Thanks for your suggestion. We have revised the conclusion.

Reviewer #2

Reviewer #2: This paper was written very well. Sea barley Hordeum marinum is an important germplasm resource. This paper suggested that tetraploid more likely originated from the diploids of H. marinum subsp. gussoneanum and the other ancestor that might be an extinct unknown diploid species. Homogenization of gene in tetraploids also occurred after polyploidization as both TRX and WAXY1 sequences in some accessions of tetraploid H. marinum subsp. gussoneanum cannot be distinguished, indicating the complicated evolution of this tetraploid. Line H824 was an interesting line, the gene was grouped into one cluster with 2X for both genes. But other 4X lines changed when they were analysed with two distinct genes. It means that the more genes used in the study, the better results will be obtained. Discuss this in the paper.

Response: Thanks for your suggestion. We added relevant discussion on this.

There are some other minor issues which need to be addressed.

Page 8:

Abstract: "H. marinum subsp. gussoneanum, diploid marinum" from two sentences, insert “and”.

Response: Thanks for your suggestion. Changes were made according to the suggestions. 

P13: Rewrite this sentence “To reveal the genetic differentiation between the diploid H. marinum subsp. Gussoneanum and the diploid H. marinum subsp. marinum used TRX, genetic analysis and neutrality test in three populations were performed.”

Response: Thanks for your suggestion. Changes were made according to the suggestions. 

P13: change 1 from “While just 1 haplotype” to one. 1 to 9 should be written out.

Response: Thanks for your suggestion. Changes were made according to the suggestions. 

Reference:

Italic Hordeum marinum for the reference 4. Double check other references.

Response: Thanks for your suggestion. We have corrected the format error of all reference.

---

## [Editor Report · Decision Letter 1]

17 Jun 2020

Phylogenetic analysis of two single-copy nuclear genes revealed origin of tetraploid barley Hordeum marinum

PONE-D-20-12057R1

Dear Dr. Ren,

We’re pleased to inform you that your manuscript has been judged scientifically suitable for publication and will be formally accepted for publication once it meets all outstanding technical requirements.

Kind regards,

Wujun Ma

Academic Editor

PLOS ONE
---

## [Editor Report · Acceptance letter]

19 Jun 2020

PONE-D-20-12057R1 

Phylogenetic analysis of two single-copy nuclear genes revealed origin of tetraploid barley *Hordeum marinum*

Dear Dr. Ren:

I'm pleased to inform you that your manuscript has been deemed suitable for publication in PLOS ONE. Congratulations! Your manuscript is now with our production department. 

Kind regards, 

on behalf of

Dr. Wujun Ma 

Academic Editor

PLOS ONE